# An Emerging Role for Type I Interferons as Critical Regulators of Blood Coagulation

**DOI:** 10.3390/cells12050778

**Published:** 2023-02-28

**Authors:** Tristram A. J. Ryan, Luke A. J. O’Neill

**Affiliations:** School of Biochemistry and Immunology, Trinity Biomedical Sciences Institute, Trinity College Dublin, D02 R590 Dublin, Ireland

**Keywords:** type I interferons, blood coagulation, IFN-α, IFN-β, thrombin, PARs, haemostasis, thrombosis, tissue factor, SLE, APS, COVID-19, cGAS-STING, neutrophil extracellular traps, FXII

## Abstract

Type I interferons (IFNs) are central mediators of anti-viral and anti-bacterial host defence. Detection of microbes by innate immune cells via pattern recognition receptors (PRRs), including Toll-like receptors (TLRs) and cGAS-STING, induces the expression of type I IFN-stimulated genes. Primarily comprising the cytokines IFN-α and IFN-β, type I IFNs act via the type I IFN receptor in an autocrine or exocrine manner to orchestrate rapid and diverse innate immune responses. Growing evidence pinpoints type I IFN signalling as a fulcrum that not only induces blood coagulation as a core feature of the inflammatory response but is also activated by components of the coagulation cascade. In this review, we describe in detail recent studies identifying the type I IFN pathway as a modulator of vascular function and thrombosis. In addition, we profile discoveries showing that thrombin signalling via protease-activated receptors (PARs), which can synergize with TLRs, regulates the host response to infection via induction of type I IFN signalling. Thus, type I IFNs can have both protective (via maintenance of haemostasis) and pathological (facilitating thrombosis) effects on inflammation and coagulation signalling. These can manifest as an increased risk of thrombotic complications in infection and in type I interferonopathies such as systemic lupus erythematosus (SLE) and STING-associated vasculopathy with onset in infancy (SAVI). We also consider the effects on coagulation of recombinant type I IFN therapies in the clinic and discuss pharmacological regulation of type I IFN signalling as a potential mechanism by which aberrant coagulation and thrombosis may be treated therapeutically.

## 1. Introduction

The type I interferon (IFN) family is expressed by most cells in humans and mice and is a critical host defence mechanism for mounting anti-viral responses. The best characterized members are IFN-α (of which there are 13 subtypes in humans and 14 in mice) and IFN-β. Type I IFNs are induced upon detection of infiltrating microbes in the bloodstream via a diverse range of interactions between pattern recognition receptors (PRRs) and pathogen-associated molecular patterns (PAMPs) or danger-associated molecular patterns (DAMPs) [1], depending on the stimulus. PRRs that induce type I IFNs include Toll-like receptors (TLRs), retinoic acid-inducible gene I (RIG-I)-like receptors, and cyclic GMP-AMP synthase (cGAS) [2]. In particular, type I IFN induction can be induced via detection of pathogen-derived dsRNA by endosomal TLR3; ssRNA by TLR7, RIG-I, and MDA-5; and cytosolic dsDNA by cGAS-STING. These are the main receptors that drive type I IFNs in response to viruses, attesting to the importance of type I IFNs in anti-viral immunity.

In addition to viral-mediated type I IFN induction, exposure to bacteria can also trigger type I IFN signalling. One well-described system of type I IFN induction in myeloid cells occurs via activation of the TLR4-mediated immune signalling pathways upon detection of endotoxin, also called lipopolysaccharide (LPS), which is found in the outer membrane of Gram-negative bacteria. Host recognition of LPS by TLR4 induces increased expression of proinflammatory cytokines in a process regulated by the nuclear factor (NF)-κB and interferon-regulatory factor (IRF) transcription factors [3]. This occurs via recruitment of myeloid differentiation primary-response gene 88 (MyD88) by MyD88-adaptor-like protein (MAL), with MyD88 forming a complex with members of the IL-1R-associated kinase (IRAK) family, in particular IRAK4, to activate NF-κB. A second signalling cascade triggered by the LPS-TLR4 interaction involves TRIF-related adaptor molecule (TRAM), which recruits TIR-domain-containing adaptor-inducing interferon-β (TRIF) to induce transcriptional upregulation of IRFs, including IRF3/7 [4]. IRFs in turn stimulate the expression of type I IFNs. This leads to IFN-α and IFN-β release, which act in an autocrine or paracrine manner via their ubiquitously expressed heterodimeric IFN-α/β receptor (IFNAR), which comprises two subunits, IFNAR1 and IFNAR2.

IFNs drive the transcription of hundreds of IFN-stimulated genes (ISGs) via the transcription factors signal transducer and activator of transcription (STAT)1, STAT2, and IRF9 of the Janus kinase (JAK)-STAT signalling pathway [5]. The precise context-dependent regulation of type I IFN induction has been reviewed in great detail previously [6,7,8,9]. Whilst type I IFNs are required early in the mounting of a host response to infection and the maintenance of homeostasis in human health and disease, overamplification of this response, or prolonged type I IFN signalling, can be detrimental [10,11]. Emerging evidence indicates that dysregulated type I IFN signalling can manifest as being a critical mediator of pathological blood coagulation in both viral and bacterial infection. Activation of coagulation may also feed-forward to amplify type I IFN production, which can have detrimental consequences. We will describe these key studies in this review and speculate on the future therapeutic implications of treating dysregulated type I IFN signalling for the management of coagulopathy.

## 2. Type I Interferons as Drivers of Blood Coagulation

Blood coagulation maintains physiological haemostasis following blood vessel injury or infection via formation of a haemostatic plug primarily comprising platelets, before activation of coagulation upon exposure of tissue factor (TF), the initiator of the extrinsic pathway of coagulation, from leukocytes or sub-endothelial tissue. TF forms a complex with coagulation factor (F)VIIa, initiating thrombin generation. Thrombin generation is then amplified and propagated in concert with activated platelets and leukocytes [12], as well as via formation of the tenase FVIIIa:FIXa and prothrombinase FXa:FVa complexes, which feed-forward to further generate thrombin. Endogenous anticoagulants such as antithrombin and tissue factor pathway inhibitor (TFPI) maintain haemostasis by rapidly inhibiting further thrombin generation, before the blood clot is dissolved via fibrinolysis [13].

Sepsis, a lethal inflammatory condition accompanied by multi-organ dysfunction, is often amplified by inflammation-induced blood vessel injury which exposes TF to be released into the bloodstream. In addition, activation of innate immune signalling pathways, such as the NLRP3 inflammasome, can also trigger TF release on extracellular vesicles from innate immune cells such as macrophages. This can occur during pyroptotic cell death during sepsis [14]. This can result in excessive thrombin generation and blood clot formation during sepsis. As a result, sepsis is often associated with coagulopathy.

Recently, numerous studies have directly implicated excessive type I IFN induction and signalling as a critical driver of blood coagulation, particularly in the context of Gram-negative bacterial-induced sepsis. This has been demonstrated using genetic mouse models lacking in core components of the type I IFN pathway. A recent study showed that *Trif*^−/−^ mice are protected from the thrombotic complications associated with LPS-induced sepsis. Following intraperitoneal LPS administration, *Trif*^−/−^ mice exhibited decreased thrombin generation and fibrin deposition in the livers and lungs compared with wild-type mice [15]. As a result, *Trif*^−/−^ mice were protected from LPS-induced lethality. Deletion of TRIF also reduced *Ifnb1* expression in the liver, spleen, and gut [15]. In the same study, investigators found that *Ifnar*^−/−^ mice, which do not express the IFN-α/β receptor, displayed decreased D-dimer levels as well as thrombin–antithrombin complex formation compared with wild-type mice following intraperitoneal LPS injection [15]. *Ifnar*^−/−^ mice also had reduced fibrin deposition in their livers and lungs compared with wild-type mice. After cecal ligation and puncture (CLP), a clinically relevant model of polymicrobial sepsis, both *Trif*^−/−^ and *Ifnar*^−/−^ mice were protected from excessive thrombin generation and coagulopathy [15]. This indicates that the TLR4-TRIF-type I IFN pathway is a critical mediator of thrombosis following Gram-negative bacterial infection.

In another study, Dejager et al. showed that serum IFN-α is significantly elevated in mice as late as 48 h post CLP-induced sepsis [16]. Treatment of wild-type mice with an anti-IFNAR1 antibody, or deletion of IFNAR, protected mice from LPS-induced lethality and CLP-induced sepsis. In addition, this also significantly reduced serum interleukin (IL)-6 levels [16]. This is notable as IL-6 (formerly known as Ifnb2) contributes to coagulation via the rapid induction and synthesis of fibrinogen in the liver [17]. In another model of CLP-induced sepsis in mice, investigators found reduced endothelial damage in *Ifnar*^−/−^ mice, with an associated suppression of inflammation-related gene expression in endothelial cells [18]. Consistent with a previous study [15], *Ifnar^−/−^* mice also displayed decreased aortic mRNA expression of *Pai-1*, an inhibitor of fibrinolysis [18], indicating that type I IFNs drive dysregulated fibrinolysis, which can lead to a prothrombotic state.

This evidence of IFNAR playing a critical role in type I IFN-mediated sepsis and coagulation is intriguing, as there is a significant similarity between IFNAR and TF, which is itself a membrane glycoprotein receptor. Global alignment of the sequence and structural patterns of IFN receptors indicates that transmembrane TF is structurally homologous to IFNAR [19]. In particular, both TF and IFNAR share evolutionarily conserved fibronectin type III domains comprising antiparallel β-sheets [20] which, in the case of TF, are proposed to be essential for the post-translational activation of TF. This process, termed decryption, occurs in the extracellular domain of TF and leads to a significant increase in TF procoagulant activity [21,22,23]. One intriguing possibility therefore is that TF might bind type I IFNs, acting as a decoy receptor or possibly a facilitator of IFNAR signalling.

### 2.1. Type I Interferon, the Caspase-11 Non-Canonical Inflammasome, and Blood Coagulation

Activation of type I IFN signalling can induce the ISG caspase-11 (in mice; caspase-4 and -5 in humans) [24], which is essential for the formation of a non-canonical inflammasome [25]. Type I IFN-mediated induction of caspase-11 occurs in response to Gram-negative—but not Gram-positive—bacteria [26]. Detection of cytosolic LPS in macrophages activates caspase-11 [27,28], which can then cleave and activate gasdermin D (GSDMD) to form pyroptotic pores in the cell membrane [29,30], through which proinflammatory and prothrombotic mediators are released. During Gram-negative bacterial sepsis, macrophage pyroptosis leads to TF release into the circulation on extracellular vesicles [31,32]. This results in excessive thrombin generation, leading to disseminated intravascular coagulation in mice [31,32]. Deletion of caspase-11 or GSDMD, or administration of an anti-TF antibody, protects mice from LPS-induced coagulation and lethality [31,32]. Caspase-11-mediated GSDMD cleavage also triggers the exposure of phosphatidylserine onto the outer membrane of macrophages [32], and this potentiates TF procoagulant activity [21,22]. Furthermore, *CASPASE-5* expression is significantly increased in primary human macrophages from sepsis patients [33]. At the transcriptional level, induction of *Gsdmd* at the mRNA level is governed by the transcription factor IRF2 [34], which lies downstream of STAT1 and STAT2. IRF2 is also essential for induction of *caspase-11* in macrophages [35], and thus IRF2 is a critical mediator of non-canonical inflammasome-induced pyroptosis. This indicates a critical role for type I IFN induction in the regulation of two key mediators of pyroptosis, which drives aberrant coagulation, and suggests that therapeutic targeting of IRF2 or the downstream non-canonical inflammasome may be a prospect for the inhibition of type I IFN-induced thrombosis.

Another key player in IFN-related coagulation is the high-mobility group box (HMGB) protein family, which is critical for host defence during both sterile and infectious injury such as sepsis, and there is an intricate interplay between type I IFN and HMGB signalling. One of these mechanisms is via cytosolic detection of DNA or RNA. In particular, HMGB1 binds to the multivalent receptor for advanced glycation end-products (RAGE) to activate the endosomal TLRs 3, 7, and 9, and thus HMGB1 is essential for cytosolic nucleic acid-mediated induction of type I IFNs [36]. As such, elevated serum HMGB1 is a biomarker of inflammatory diseases [37]. Upon innate immune cell activation, HMGB1 translocates from the nucleus to the cytosol. This occurs upon activation of JAK/STAT1 via type I IFN signalling [37]. HMGB1 is then released from immune cells via inflammasome-mediated pyroptosis, which is induced upon autophosphorylation of the dsRNA-dependent protein kinase R (PKR), which is itself an ISG [38]. This results in physical interaction between PKR and the inflammasomes (NLRPs 1 and 3, NLRC4, and AIM2) [39]. Extracellular HMGB1 can then feed-back to further amplify the type I IFN signalling cascade in macrophages via activation of TLR4 [40]. In addition, HMGB1 physically binds extracellular LPS and is internalized into macrophage lysosomes via RAGE [41]. Destabilization of the lysosomal membrane by HMGB1 then releases LPS into the cytosol of macrophages, where it cleaves and activates caspase-11 to mediate pyroptosis, resulting in the release of proinflammatory and prothrombotic cytokines including IL-1β, IL-18, TF, and further HMGB1, amplifying and exacerbating the resulting inflammation [41]. Therefore, HMGB1 activity is critical for feed-forward inflammation and aberrant blood coagulation, possibly via type I IFNs. In addition, HMGB1 may contribute to blood coagulation by inducing *F3* (TF) mRNA and TF protein, as well as TF procoagulant activity, in macrophages and endothelial cells [42]. This is likely due in part to HMGB1 being able to induce phosphatidylserine exposure from macrophages in a pyroptosis-dependent manner [15]. Induction of *F3* by HMGB1 may also occur via activation of the transcription factor NF-κB [42], which is a transcriptional regulator of *F3* [43]. Furthermore, platelet-derived HMGB1 has been implicated as an important contributor to neutrophil extracellular trap (NET) formation and subsequent deep vein thrombosis in mice [44]. These studies describing type I IFNs as drivers of blood coagulation are depicted in Figure 1.

A recent study also implicated interferon-inducible transmembrane (IFITM) proteins, which block the early stages of viral replication, as drivers of platelet activation in bacterial sepsis. In vitro stimulation of megakaryocytes with IFN-α induced IFITM3 protein expression via STAT1 phosphorylation, as well as mTOR [45]. This increased fibrinogen endocytosis in megakaryocytes via localization of the integrin α_IIb_β_3_ and clathrin into lipid rafts. This translated to an increase in platelet aggregation in mice in vivo. Furthermore, IFITM3 expression and fibrinogen endocytosis were increased in platelets from humans with non-viral sepsis [45]. Therefore, IFITMs are required for IFN-α-induced thrombosis.

### 2.2. cGAS-STING and Blood Coagulation

Mounting evidence indicates a critical role for cGAS-STING signalling in the regulation of blood coagulation following infection. cGAS senses cytosolic dsDNA, generating the second messenger cyclic guanosine monophosphate–adenosine monophosphate (cGAMP), which activates STING. This promotes activation of TANK-binding kinase 1 (TBK1), which in turn phosphorylates IRF3 [46], resulting in type I IFN production [47]. The severe autoinflammatory syndrome, STING-associated vasculopathy with onset in infancy (SAVI), is a rare clinical condition where patients with gain-of-function mutations in *TMEM173* (the gene that encodes for STING) present with cutaneous vasculopathy and vasculitis [48,49]. Severe acute respiratory syndrome coronavirus 2 (SARS-CoV-2) can also induce cGAS-STING activation. This occurs via the release of mitochondrial DNA (mtDNA) into the cytosol, which is detected by cGAS [50]. Following SARS-CoV-2 infection, cGAS-STING activates NF-κB-mediated inflammatory cytokine production as well as TBK1-IRF3-mediated type I IFN induction [51] to sustain the anti-viral response. In the context of blood coagulation, SARS-CoV-2-infected human endothelial cells exhibit increased *IFNB1* and *F3* mRNA and decreased *TFPI* mRNA expression, which are restored to basal levels upon treatment with the STING inhibitor H-151 [50]. H-151 also significantly reduced the SARS-CoV-2-induced expression of *IL6* [50], which is prothrombotic [17]. Therefore, cGAS-STING likely contributes to type I IFN induction and COVID-associated pathology, including coagulopathy. However, the potent induction of type I IFNs by cGAS-STING can be harnessed to promote the host anti-viral response to SARS-CoV-2 infection. Humphries et al. found that a pharmacological STING activator, diABZI-4, potently induced type I IFNs and suppressed SARS-CoV-2 replication in human A549 lung epithelial cells [52]. Therapeutic administration of the STING agonist protected mice from SARS-CoV-2-induced pulmonary damage and mortality, with direct STING activation proving more effective at eliminating weight loss and reducing mortality than the administration of IFNs in mice after SARS-CoV-2 infection [52]. This raises the possibility of targeting STING for the treatment of COVID-19-associated inflammation and coagulopathy.

STING may also drive blood coagulation following infection in a type I IFN-independent manner. In one study, STING increased GSDMD-mediated TF release in monocytes and macrophages [53]. The authors, however, noted that this occurred in a type I IFN-independent manner as there were no differences in disseminated intravascular coagulation (DIC) markers, including fibrinogen, D-dimer, fibrin, and TF release in the plasma of *Ifnar*^−/−^ mice compared with wild-type mice 48 h after CLP-induced sepsis. The acute nature of type I IFN production in the innate immune response and in driving blood coagulation might explain why the authors attributed the role of STING in these models to be type I IFN-independent. It may also be because cGAS-STING can trigger pyroptosis in a type I IFN-independent manner. Gaidt et al. showed that the detection of cytosolic DNA results in pyroptosis via activation of TBK1 and IKKε to induce type I IFN-independent NLRP3 activation, as well as simultaneous type I IFN-dependent induction of caspase-11 [54]. Mechanistically, STING promoted DIC via binding to ITPR1 on the endoplasmic reticulum (ER) to promote ER calcium release, which decrypts TF [22,55] and triggers TF release via GSDMD-induced pyroptotic pores [53]. Therefore, the therapeutic targeting of STING may have beneficial effects for the treatment of inflammation-associated coagulopathy, regardless of the extent of the contribution of type I IFNs in those conditions.

### 2.3. Type I Interferon, COVID-19, and Blood Coagulation

The host IFN response to viral infection has been the focal point of much research in the past three years, as it is an essential defence mechanism following SARS-CoV-2 infection. In particular, rapid type I IFN induction is necessary for the mounting of an effective anti-viral response against COVID-19. However, aberrant type I IFN signalling in COVID-19 is detrimental either due to overactivation of type I IFNs, or via SARS-CoV-2-mediated disruption of cellular RNA splicing and translation, as well as degradation of host mRNAs, which limits ISG production and enables further propagation of SARS-CoV-2 [56,57]. Thus, COVID-19 pathology is associated with dysregulated type I IFN signalling [58,59,60]. COVID-19 is also characterized by systemic coagulopathy [61]. This includes elevated plasma levels of D-dimer in patients [62] and increased pulmonary deposition of the thrombotic marker fibrinogen/fibrin [63], as well as elevated pulmonary von Willebrand Factor (vWF) deposition, which is a clinical marker of both acute and sustained endothelial cell activation, following SARS-CoV-2 infection [64]. Innate immune cells are core contributors to this pathology, with a recent report identifying a transcriptional shift in monocytes to a more prothrombotic genotype following SARS-CoV-2 infection [65]. Expression of *F3* and TF-positive microvesicles are also increased in monocytes, macrophages, and platelets [66], as well as in endothelial cells and epithelial cells from patients with severe COVID-19 [50,67], propagating the coagulopathy associated with COVID-19. TF-positive microvesicles then drive excessive thrombin generation and coagulopathy following infection with SARS-CoV-2 [68,69]. In addition, thrombin and FXa have recently been shown to directly cleave the SARS-CoV-2 spike protein, augmenting viral entry into human airway epithelial cells and human pluripotent stem cell-derived lung organoids [70]. This suggests a broader anti-viral effect of deploying anticoagulants, in particular direct thrombin inhibitors and direct FXa inhibitors, as well as inhibitors of type I IFNs, in the treatment of COVID-19, as this will likely suppress pathological type I IFN production (as a result of reduced viral uptake) as well as thromboinflammation.

Assessment of whole blood from COVID-19 patients found a correlative relationship between defective type I IFN signalling, elevated coagulation markers, and increasing disease severity [60]. In addition, macrophage type I IFN- and caspase-11-dependent pyroptosis have been implicated in mediating TF-dependent coagulopathy in a mouse model of COVID-19 [71]. Therefore, investigators have attempted to target this type I IFN-caspase-11–TF axis as a means of limiting inflammation and coagulopathy in COVID-19. Heparin, the clinically approved antithrombin activator, can also block caspase-11-mediated pyroptosis at concentrations lower than those required for heparin to activate antithrombin [72]. Notably, anticoagulation treatment with heparin has been shown to decrease mortality in non-severe COVID-19 patients [73] and patients with elevated D-dimer levels [74]. However, a caveat of using heparin for the treatment of COVID-19 is that it is not effective in severe patients when administered therapeutically versus standard care pharmacologic thromboprophylaxis [75]. This indicates a correlation between the rapid type I IFN response and the efficacy of heparin therapy following SARS-CoV-2 infection, with heparin proving most effective when administered prior to the onset of severe pathology. Therefore, inhibition of type I IFNs and the non-canonical inflammasome is an attractive target for anticoagulation treatment in COVID-19 associated coagulopathy, as well as during bacterial infections.

Emerging evidence indicates that the presence of persistent fibrin amyloid microclots may provide a mechanistic basis for the long-term effects of long COVID (also termed post-acute sequelae of COVID-19), which include fatigue and vertigo, as fibrin amyloid microclots block capillaries and the transport of oxygen to tissues [76,77]. Type I IFNs may contribute to the development of microclots in long COVID as elevated type I IFN expression has been detected for at least 8 months after infection [78]. Thus, new studies are urgently required to study the effect of IFN-mediated coagulation in long COVID.

### 2.4. Type I Interferonopathies and Blood Coagulation

Type I interferonopathies, an umbrella term first coined in 2011 by Yanick Crow [79], describes a group of clinical conditions associated with sustained, elevated type I IFN production. These related syndromes include systemic lupus erythematosus (SLE), an autoimmune disease characterized by persistent type I IFN upregulation [10]. Patients with SLE are at significantly greater risk of developing atherothrombotic cardiovascular disease [80]. PBMCs from patients with SLE have elevated mRNA expression of *F3* [81]. SLE patients also have a greater number of activated platelets compared with healthy controls, and these platelets exhibit an elevated type I IFN mRNA and protein signature [82]. In addition, SLE patients with a history of vascular disease have increased type I IFN-regulated protein levels compared with SLE patients without a history of vascular disease [82]. Furthermore, in SLE patients, IFN-α rapidly triggers apoptosis of endothelial progenitor cells (EPCs) and myelomonocytic circulating angiogenic cells (CACs), which are required for blood vessel repair [83]. SLE EPCs and CACs exhibit increased IFN-α expression and an elevated type I IFN signature [83]. This indicates that sustained type I IFN production in SLE is closely intertwined with vascular damage and coagulopathy. This was demonstrated in one study whereby deletion of IFNAR in lupus-prone mice improved endothelial function and decreased atherosclerosis severity [80]. One primary mechanism of type I IFN induction in lupus occurs via the release of oxidized mtDNA, which is elevated in skin lesions from lupus patients and is associated with increased type I IFN production [84]. As cGAS-STING detects dsDNA and is a potent inducer of type I IFNs, cGAS-STING likely plays a key role in the pathogenesis of SLE. This is demonstrated by elevated *cGAS* expression in PBMCs from SLE patients [85]. Furthermore, expression of apoptosis-derived membrane vesicles, which are associated with elevated dsDNA levels, have been shown to activate cGAS-STING to induce type I IFNs in serum from SLE patients [86].

In addition to the thrombotic complications associated with SLE and SAVI, the importance of a functional type I IFN signalling pathway for regulating innate immune pathways and control of blood coagulation is demonstrated by the mutation of core signalling components which can predispose to increased risk of mortality. For example, a mutation in JAK2 first identified in 2005, JAK2V617F, is present in >80% patients with polycythaemia vera, a myeloproliferative leukaemia whereby excessive erythrocyte production can lead to excessive blood clotting and is associated with increased mortality [87]. In addition, the JAK2V617F mutation can also lead to essential thrombocythemia, a neoplasm whereby increased platelet production may further propagate the risk of excessive blood clotting [88]. Although the exact mechanisms underlying the contribution of type I IFNs to the increased thrombosis risk in these clinical conditions remains to be elucidated, this further indicates that dysregulated type I IFN signalling is a critical signal that drives aberrant blood coagulation.

### 2.5. Pharmacological Targeting of Type I Interferons to Treat Coagulopathies

The association between dysregulated type I IFN signalling and coagulopathy was highlighted during the COVID-19 pandemic, with pharmacological inhibition of excessive type I IFN production being associated with a concomitant reduction in SARS-CoV-2-induced coagulopathy. A number of studies reported the beneficial effects of inhibition of type I IFN signalling, particularly using JAK inhibitors, which are clinically approved for the treatment of conditions such as rheumatoid arthritis and myeloproliferative neoplasms (MPNs). For example, the JAK1/3 inhibitor tofacitinib lowered the SARS-CoV-2-induced risk of death or respiratory failure over 28 days versus placebo [89]. Another clinical trial found a decrease in D-dimer and C-reactive protein levels in the blood of hospitalized SARS-CoV-2 patients when treated with the JAK1/2 inhibitor baricitinib plus corticosteroids, versus treatment with corticosteroids alone [90]. JAK inhibitors have been especially effective in COVID-19 patients receiving high-flow oxygen or non-invasive ventilation [91], indicating that pharmacological inhibition of type I IFN signalling is most beneficial prior to the onset of severe disease. JAK inhibition will also block IL-6 signalling, further limiting inflammation and coagulation. Increased TYK2 expression is also associated with mortality in COVID-19 patients [92]. Therefore, targeting JAKs with specific inhibitors and therefore downstream JAK-TYK signalling may confer protection on COVID-19 patients. In addition, JAK inhibition may also suppress thromboinflammation in COVID-19.

## 3. The Effect of Blood Clotting on Type I Interferons

There is also a growing body of evidence which suggests that the key mediators of blood clotting, the coagulation factors themselves, can act on innate immune cells to induce proinflammatory cytokines and type I IFNs. This can amplify IFN production to combat bacterial or viral infection and restore haemostasis via the rapid resolution of inflammation, but it can also trigger a detrimental, pathological inflammatory cycle, as we will describe below.

The heterotrimeric GTP-binding protein-coupled protease-activated receptors (PARs) are expressed by a range of immune cells, and whilst PAR signalling is essential for the maintenance of haemostasis, it can also lead to the induction of proinflammatory cytokines [93]. PARs are the main substrate for thrombin and therefore numerous studies have assessed the induction of type I IFNs via thrombin-PAR signalling. The core procoagulant role of thrombin is the cleavage of fibrinogen into fibrin to generate a thrombus by forming a mesh at the sites of infection and vascular damage, in conjunction with activated platelets and neutrophils which expel their DNA, histones, and granule-derived enzymes during NETosis. Thrombin generation can occur as the endpoint of the intrinsic/contact (FXII-mediated) or extrinsic (TF-mediated) pathways of the coagulation cascade. Moreover, it has recently emerged that extracellular vesicles on erythrocytes may also contribute to the generation of thrombin [94,95]. However, excess thrombin generation can be pathological in a range of clinical conditions, resulting in tissue ischaemia by microvascular and macrovascular thrombosis.

### PAR-Mediated Induction of Type I Interferons

Thrombin can trigger inflammatory signalling through PARs, which can result in a process termed thromboinflammation. PARs 1, 3, and 4 recognize and are cleaved and activated by thrombin [93]. In addition, a recent study suggests that thrombin, when bound to the endogenous anticoagulant thrombomodulin, can also cleave PAR2 [96]. Thrombin-PAR signalling is critical for the interplay between inflammation and coagulation, boosting proinflammatory cytokine secretion, as PARs can physically interact and therefore synergize with TLRs on innate immune cells. For example, Subramaniam et al. found that although stimulation of human umbilical vein endothelial cells (HUVECs) with thrombin did not directly induce the mRNA expression of ISGs or *F3*, co-stimulation of HUVECs with both thrombin and the dsRNA polyinosinic:polycytidylic acid (poly(I:C)) resulted in significantly increased expression of *F3* and TF procoagulant activity compared with poly(I:C) stimulation alone [97]. This provides evidence of PAR1/2 and TLR3 synergy and suggests that in the context of an innate immune response, thrombin may positively feed-back to amplify its own production.

Recent studies have employed PAR knockout mice to examine further the effects of thrombin signalling on the innate immune system. Macrophages and splenocytes from *Par1*^−/−^ mice exhibited decreased type I IFN signalling after administration of poly(I:C) [98]. Furthermore, mRNA expression of *Ifnb1*, *Irf7*, and *Cxcl10* were reduced in *Par1*^−/−^ mice after infection with Coxsackievirus group B (CVB), suggesting that PAR1, and therefore thrombin signalling, contributes to type I IFN-mediated anti-viral responses [98]. Furthermore, after stimulation of mouse cardiac fibroblasts with poly(I:C), PAR1 and TLR3 synergized to drive an anti-viral but proinflammatory response via induction of IFN-β and CXCL10 via increased phosphorylation of the MAPK p38 [99]. In vivo, poly(I:C) administration increased expression of *F3* in the heart and liver as well as thrombin generation in mouse plasma. Administration of an anti-TF monoclonal antibody or the thrombin inhibitor dabigatran etexilate significantly increased CVB3-induced myocarditis [99], indicating that haemostatic coagulation contributes to the innate, anti-viral response. Moreover, the chemotherapeutic drug, doxorubicin, has recently been found to increase thrombin generation in a TF-dependent manner, driving thromboinflammation in mice via PAR1 activation in cardiomyocytes and cardiac fibroblasts [100]. This prothrombotic phenotype may explain the underlying basis for the cardiotoxicity associated with doxorubicin.

In addition, PAR2 has also been shown to synergize and physically interact with TLRs during inflammation [101]. PAR2 synergized with TLRs 2, 3, and 4 in mucosal epithelial cells following poly(I:C) stimulation, which activated NF-κB via degradation of IκBα and phosphorylation of p65 [102]. However, in contrast to PAR1, PAR2 negatively regulated the TLR3 signalling pathway, resulting in decreased phosphorylation of IRF3 & STAT1, and therefore suppressing the type I IFN-mediated anti-viral response. Thus, *Par2*^−/−^ and *Tlr4*^−/−^ mice were protected from lethality induced by infection with H1N1 influenza A virus [102]. Additionally, in LPS-stimulated bone marrow-derived macrophages, PAR2 activation resulted in increased IL-10 secretion, possibly via increased STAT3 phosphorylation, but decreased secretion of the proinflammatory cytokines IL-6, TNF, and IL-12p40 [103]. This suggests that PAR2 counteracts LPS-induced proinflammatory cytokine production. Furthermore, PAR2 restrained type I IFN signalling in fibroblasts via the binding of TLR3 in a mouse model of CVB3-induced myocarditis [104]. Higher cardiac *PAR2* mRNA expression correlated with low *IFNB1* expression in patients with non-ischaemic cardiomyopathy, resulting in increased expression of inflammatory markers, including CD3+ and CD45+ T cells [104]. Thus, modulation of the host response by PAR2 differs in response to activation of different TLRs. Therefore, synergy of individual PARs with TLRs can have contrasting effects on downstream innate immune signalling, particularly with regards to type I IFN signalling.

In addition to the induction of type I IFNs by thrombin-PAR-TLR signalling, thrombin can also induce the proinflammatory cytokines IL-6, IL-1β, and TNF in human monocytes [105] and vascular smooth muscle cells [106]. Intraperitoneal injection of thrombin in mice increased IL-6 secretion from peritoneal macrophages into the peritoneum in a fibrinogen-dependent manner [107]. Thrombin can also cleave and activate pro-IL-1α (p33) into its active form (p18) when expressed on the surface of macrophages, platelets, and keratinocytes [108]. Thrombin-activated IL-1α was found to be important for rapid thrombopoiesis and wound healing. Furthermore, thrombin-cleaved IL-1α is elevated in the plasma of ARDS patients versus healthy controls [108], and thus can be considered a biomarker of thromboinflammatory conditions.

Furthermore, the formation of TF:FVIIa rapidly induces FXa to drive cytokine production via PAR2 [109]. TF:FVIIa:FXa can also form a complex with EPCR to trigger TLR4/PAR2-mediated type I IFN signalling. This occurs via induction of pellino-1, the TLR3/4 adaptor protein, in addition to IRF8 [110]. Deficiency of EPCR, PAR2, or TF in mice attenuates LPS-induced expression of IRF8 and subsequent type I IFN induction in vitro and in vivo [110]. This indicates a role for TF:FVIIa as a DAMP by activating innate immune signalling pathways. The role of PARs in the induction of type I IFNs and downstream blood coagulation is summarized in Figure 2.

## 4. Interferon Therapy and Coagulopathy

In the clinic, type I IFNs are administered therapeutically. Recombinant IFN-α is approved for the treatment of chronic hepatitis B and C viral infections as well as various cancers, including MPNs, whilst recombinant IFN-β is approved for multiple sclerosis (MS) treatment to regulate the persistent inflammation associated with the condition. Although these therapies are effective, there have been indications that IFN therapy can lead to an increased risk of thrombosis.

Elevated vWF antigen expression and activity has been detected in the plasma of MPN patients receiving IFN-α compared with healthy controls [111]. Plasma from IFN-α-treated MPN patients also displayed significantly increased activity of fibrinogen and the coagulation factor FVIII, as well as reduced protein S activity, indicating a shift in MPN patients to a more procoagulant phenotype. Functionally, this resulted in elevated thrombin generation in MPN patient plasma. The investigators tracked the patients for 6 months and found that haemostasis was restored in patients when IFN-α treatment was discontinued, as demonstrated by a significant reduction in vWF and fibrinogen levels, as well as increased protein S activity [111]. Thus, IFN-α therapy increases prothrombotic biomarkers in the plasma of MPN patients. Recombinant type I IFN therapies have also been linked with a dose-dependent increase in the risk of thrombotic microangiopathies in MS patients [112]. A study by Jia et al. suggested a potential mechanism by which type I IFNs might drive thrombotic microangiopathies. They compared the effects of recombinant IFN-α and IFN-β on endothelial cell function and found that IFN-β suppressed proliferation and survival of HUVECs, but IFN-α did not affect these parameters [113]. Meanwhile, both IFN-α and IFN-β blocked angiogenesis via activation of IFN-inducible CXCL10 when HUVECs and human dermal fibroblasts were co-incubated in vitro. Endothelial cell activity was impaired by IFN-α and IFN-β via inhibition of endothelial cell-produced nitric oxide and prostacyclin. Intriguingly, IFN-β significantly increased PAI-1 and downregulated uPA in HUVECs, which is indicative of decreased fibrinolysis [113]. Thus, these studies indicate that type I interferonopathies or administration of type I IFNs may be associated with increased risk of pathological blood clotting.

## 5. The Interrelationship between Type I Interferons and Thrombosis in Disease

In addition to the emerging evidence linking aberrant type I IFN signalling with coagulopathy in conditions such as Gram-negative bacterial infection and COVID-19, type I interferonopathies such as SLE have been associated with an increased risk of thrombosis, as discussed above. Emerging evidence indicates a key role for excessive neutrophil activation and NET formation in the pathogenesis of SLE. Activation of neutrophils with ribonucleoprotein immune complexes, which are highly expressed in lupus, induce hyperpolarization of mitochondria, followed by the translocation of mitochondria to the cell surface and subsequent release of mtDNA [114]. Oxidized mtDNA drives NETosis in SLE and lupus-like diseases [114], leading to increased deposition of dsDNA, IL-17, HMGB1, and the anti-microbial peptide LL-37 in NETs from SLE patients [115,116]. LL-37 induces type I IFNs in plasmacytoid dendritic cells (pDCs) by binding extracellular dsDNA and transporting it into endosomal compartments of pDCs, triggering TLR9-mediated IFN production [117]. LL-37 may also transport dsDNA into monocytes to induce type I IFN signalling via cytosolic STING-TBK1 activation [118]. Interestingly, this occurs independently of TLR9 in monocytes [118]. This is supported by the fact that stimulation of human PBMCs with NETs induces *IFNA1* and *IFNB1* expression [119]. Thus, NETs are interferogenic. However, NETs are also prothrombotic, particularly in COVID-19 [120], in part by capturing TF and TF-positive extracellular vesicles from the circulation [121,122], thereby facilitating activation of the extrinsic pathway of coagulation. Furthermore, whilst neutrophil-released DNA and histones are prothrombotic, intact NETs are not directly thrombogenic themselves [123]. This is notable as DNA released from NETs can drive thrombin generation in a FXII-dependent manner [124], with a FXII-NETs cross-talk implicated in the pathogenesis of COVID-19 [125] as well as deep vein thrombosis [126]. In addition, in a feed-forward manner, FXII has been shown to drive NETs in a baboon model of *Escherichia coli*-induced sepsis [127]. Increased levels of autoantibodies to FXII have also been associated with thrombosis in SLE [128], suggesting the possibility that FXII might drive type I IFN signalling in SLE via NET formation. Future studies should assess this hypothesis to potentially unravel FXII-mediated thrombosis as a novel target for SLE therapy.

The autoimmune disease antiphospholipid syndrome (APS) is also associated with an increased risk of aberrant type I IFN signalling and thromboinflammation. In APS, endocytosis of antiphospholipid antibodies (aPLs) into pDCs induces TLR7/8-dependent type I IFN production [129], as characterized by increased expression of *TLR7* [130] and *TLR8* [131] in PBMCs from patients with APS. This leads to IFN-α release from pDCs, which stimulates the production of B1a cells, a type of B cell associated with autoimmune diseases such as APS and SLE. B1a cells then produce further lipid-reactive aPLs, propagating APS [129]. aPLs also hijack haemostatic control of coagulation by inhibiting TFPI [132]. This results in increased thrombosis via TF:FVIIa-dependent thrombin generation, highlighted by elevated *F3* expression in monocytes and PBMCs in patients with APS [133,134], demonstrating the prothrombotic genotype associated with APS. Furthermore, monocytes from patients with APS and SLE exhibit increased expression of *PLSCR1* [135], which is involved in monocyte phosphatidylserine externalization and TF decryption [23]. In the presence of an aPL, stimulation of macrophages with IFN-α significantly increased *F3* expression [135], indicating that type I IFN signalling drives TF-dependent thrombin generation in APS. Soluble TF levels are elevated in plasma from patients with APS [136], and there is an increase in TF-dependent procoagulant activity in the PBMCs of patients with APS [137] as well as in the carotid artery homogenates of mice injected intraperitoneally with serum IgG isolated from APS patients [138]. This suggests a mechanism (and potential therapeutic target) for the innate immune-mediated microthrombosis associated with APS. In addition, identifying the molecular mechanisms underlying a prothrombotic type I IFN–TF axis in APS would contribute greater understanding of the complex processes driving thrombosis in APS.

Furthermore, excessive type I IFN production via cGAS-STING activation has been implicated in further models of inflammation-associated coagulopathy, including cerebral venous sinus thrombosis [139] and acute lung injury [140], indicating a broad spectrum of conditions where there is a strong correlation between dysregulated type I IFN production and aberrant coagulation. It would be intriguing to hypothesize that these two events are interrelated, and thus future studies should assess the extent of the cross-talk between type I IFN and thrombosis in the pathogenesis of these clinical conditions.

## 6. Conclusions

Mounting evidence indicates that dysregulated host type I IFN production can manifest as being a critical driver of pathological blood coagulation during infection but also in such conditions as SLE and SAVI, and possibly during IFN therapy. Type I IFNs can trigger pathological coagulation via both the intrinsic/contact and extrinsic pathways of blood clotting. Coagulation factors can feed-forward to induce proinflammatory cytokines and type I IFNs, for example via thrombin-PAR-TLR signalling. Therefore, type I IFNs may be critical drivers of thrombosis. With this review, we hope to inspire greater focus on the precise mechanisms by which type I IFNs mediate blood coagulation and thrombosis, and vice versa, which may prove that type I IFNs lie at the fulcrum of inflammation and coagulation. Targeting IFNs may therefore present therapeutic opportunities for the treatment of aberrant coagulation in infection and inflammatory diseases, as well as providing new avenues for the prevention or treatment of DIC in Gram-negative bacterial sepsis.

## Figures and Tables

**Figure 1 cells-12-00778-f001:**
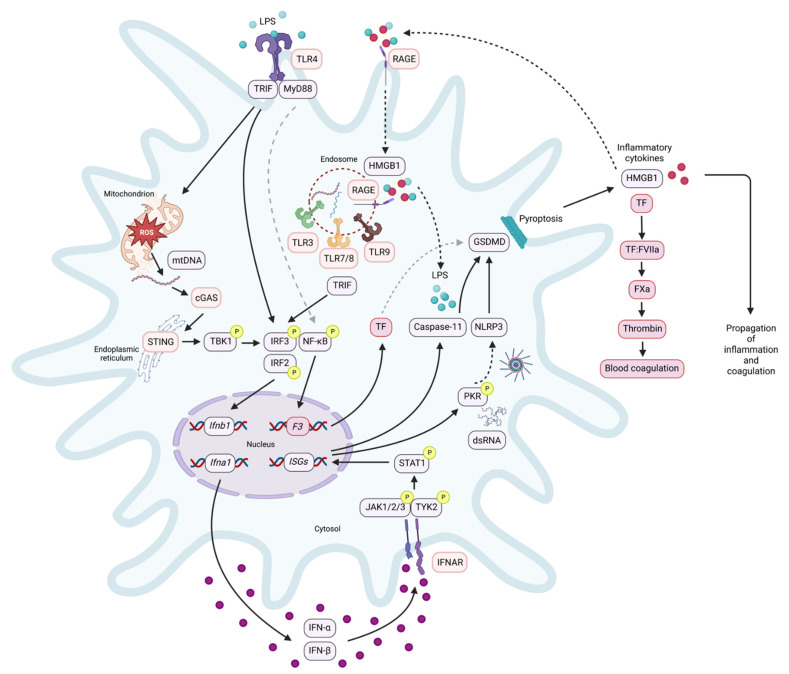
Type I interferons as drivers of blood coagulation. Activation of innate immune cells occurs upon detection of PAMPs (such as LPS) by PRRs (such as TLR4). This rapidly induces type I IFNs via TRIF or NF-κB. The cytosolic PRR cGAS-STING can also induce type I IFNs upon detection of dsDNA, which can be released from hyperpolarized mitochondria. This results in the release of IFN-α and IFN-β, which act via their receptor IFNAR to induce hundreds of ISGs. Whilst this is a potent and necessary host anti-viral response, excessive type I IFN production can trigger excessive blood coagulation, particularly following Gram-negative bacterial infection. Caspase-11 and PKR, which are ISGs, trigger pyroptosis via inflammasome activation. This releases proinflammatory and procoagulant cytokines, including TF, the initiator of the extrinsic pathway of coagulation. HMGB1 is also released and can physically bind with LPS to activate RAGE. Internalization of HMGB1-LPS may trigger further induction of type I IFNs via activation of endosomal TLRs, potentially amplifying the associated inflammation and coagulation.

**Figure 2 cells-12-00778-f002:**
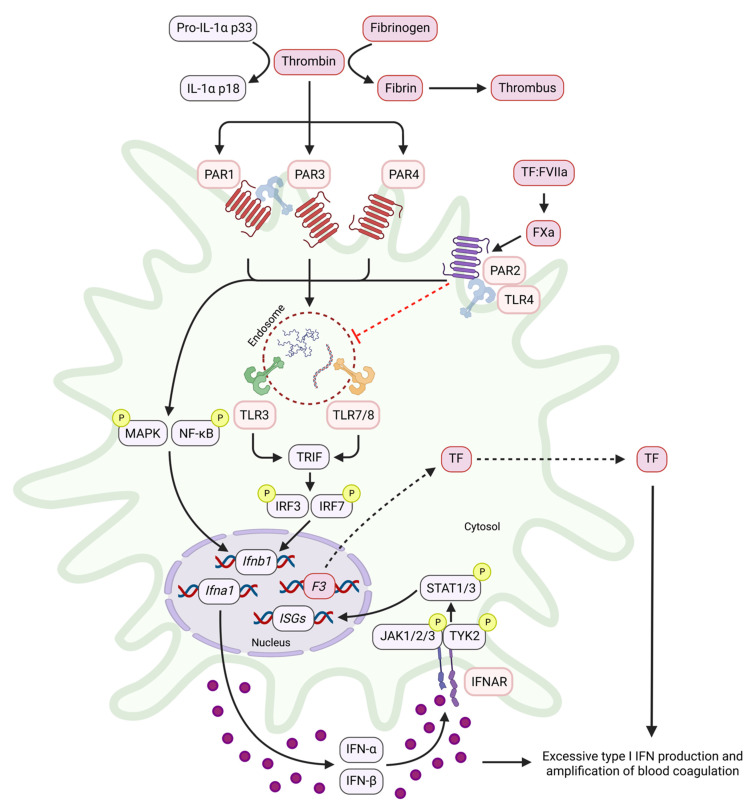
Induction of type I IFNs and downstream blood coagulation by PARs. In addition to thrombin’s procoagulant (cleavage of fibrinogen) and proinflammatory (cleavage of IL-1α) roles, thrombin can also activate PARs 1, 3, and 4, which can synergize with TLRs to induce type I IFNs. This can occur either via the NF-κB or TRIF signalling cascades, resulting in the production of IFN-α and IFN-β, which act via IFNAR to induce anti-viral but proinflammatory ISGs. PAR2, however, which is activated by FXa via TF:FVIIa complex formation, suppresses type I IFN induction whilst activating NF-κB. This results in the induction of *F3*, leading to TF release and the further feed-forward amplification of PAR2-TLR signalling. Thus, activation of PARs can trigger type I IFN production, which can amplify the inflammatory and procoagulant response during infection.

## Data Availability

Not applicable.

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
