# Peer review of "An Emerging Role for Type I Interferons as Critical Regulators of Blood Coagulation"

_cells, 2023, doi:10.3390/cells12050778_

Round 1

Reviewer 1 Report

The present paper by Ryan and O’Neill reviews existing literature on the bidirectional interactions between type 1 interferons (INF) and blood coagulation and their potential contribution to thromboinflammatory conditions. The paper is well written and summarizes the literature nicely. I have some specific comments for the authors as detailed below:

1) Minor comment: A typo in the heading “1. Introduction”

2) In the section “2. Type 1 interferons as drivers of blood coagulation” a short description of the haemostatic system could be given for the benefit of the reader who is not so familiar with this system. This could be based on the cell-based model of haemostasis with the initiation (TF-driven), amplification and propagation phases (formerly the extrinsic, intrinsic and common pathways).

3) Page 3, lines 102-104: “In addition, this also significantly reduced serum interleukin (IL)-6 levels [16]. This is notable as IL-6 (formerly known as Ifnb2) contributes to coagulation via rapid induction and synthesis of fibrinogen in the liver [15].” It is true that IL-6 induces fibrinogen synthesis in the liver as part of the multi-faceted acute phase response, but a high plasma-fibrinogen in itself does not make the blood clot.

4) Page 3, lines 104-107: “In another model of CLP-induced sepsis in mice, investigators found reduced endothelial damage in Ifnar-/- mice, with an associated decrease in fibrinolysis and suppression of inflammation-related gene expression in endothelial cells [16].” As I read this study, the authors did not actually measure fibrinolysis, but they found reduced PAI-1 gene expression in the endothelial cells. PAI-1 is an inhibitor of fibrinolysis, so decreased PAI-1 expression should theoretically increase fibrinolysis.

5) Page 3, lines 116-117. One intriguing possibility therefore is that TF might bind type I IFNs, …” This is a possibility, but it appears a bit speculative. Is there any evidence for this?

6) Page 3, lines 125-127: “During gram-negative bacterial sepsis, macrophage pyroptosis leads to TF release into the circulation on extracellular vesicles [29, 30]. This results in excessive thrombin generation leading to disseminated intravascular coagulation in mice.” Please clarify which studies the latter statement is based on. Is is also references 29-30?

7) Page 4, lines 162-164: “In addition, HMGB1 may con- 162 tribute to blood coagulation by inducing F3 (TF) mRNA and TF protein, as well as TF procoagulant activity, in macrophages and endothelial cells.” Please provide the relevant reference for this statement?

8) Page 7, lines 263-267: “This suggests a broader anti-viral effect of deploying anticoagulants, in particular direct thrombin inhibitors and direct FXa inhibitors, as well as inhibitors of type I IFNs, in the treatment of COVID-19, as this will likely suppress pathological type I IFN production (as a result of reduced viral uptake) as well as thromboinflammation.” Is there any evidence to suggest a net beneficial effect of treatment with direct thrombin inhibitors or direct FXa inhibitors in COVID-19?

9) Page 7, lines 274-275: “Heparin, the clinically approved thrombin inhibitor, can also block caspase-11-mediated pyroptosis.” Heparins are not the only clinically approved thrombin inhibitors. Direct thrombin inhibitors (which you mention above) such as dabigatran etexilate and argatroban are also clinically approved. Nor are heparins specific thrombin inhibitors as they also inhibit FXa. Also it could be specified which type of heparin are meant.

10) Page 7, lines 277-278: “However, a caveat of using heparin for the treatment of COVID-19 is that it is not effective in severe patients [73].” The paper that you references compared therapeutic-dose low molecular weight heparin (LMWH) to prophylactic- or intermediate-dose LMWH, i.e. it did not compare LMWH to no LMWH.

11) Page 7, lines 287-289: “Type I IFNs may contribute to the development of microclots in long COVID as elevated type I IFN expression has been detected for at least 8 months after infection [76].” This is possible but does appear a bit speculative.

12) On page 8, lines 317-327, you write: “In addition to the thrombotic complications associated with SLE and SAVI, the importance of a functional type I IFN signalling pathway for regulating innate immune pathways and control of blood coagulation is demonstrated by mutation of core signalling components which can predispose to increased risk of mortality. For example, a mutation in JAK2 first identified in 2005, JAK2V617F, is present in >80% patients with polycythaemia vera, a myeloproliferative leukaemia whereby excessive erythrocyte productioncan lead to excessive blood clotting and is associated with increased mortality [85]. In addition, the JAK2V617F mutation can also lead to essential thrombocythemia, a neoplasm whereby increased platelet production may further propagate the risk of excessive blood clotting [86].” Based on this, you conclude: “Collectively, these clinical conditions provide compelling evidence that a functional type I IFN signalling pathway is a critical regulator of blood coagulation.” I really don’t think that you can say that. PV and ET are not coagulopathies in a traditional sense. It is true that PV and ET are associated with increased thrombosis risk, and some of the mechanisms behind still wait to be elucidated including the contributing of inflammation, but the most important driver behind the thrombosis risk in these conditions is the increased cell count (erythrocytes, leukocytes and/or platelets) which leads to both hyperviscosity and closer interactions between platelets and endothelium, as well as increased platelet activation. And cytoreduction is shown to alleviate thrombosis risk.

13) Page 8, line 342, regarding treatment with tofacitinib and baricitinib: “JAK inhibition will also block IL-6 signalling, further limiting inflammation and coagulation.” Can you then say, from the studies that you refer, whether it is IFN or IL-6 blocking that gives the effect?

14) Page 8, line 357: “PARs are the main substrate for thrombin…”. One could also argue that fibrinogen is the main substrate for thrombin. But PARs are certainly important thrombin-activated receptors.

15) Page 9, lines 358-360: “The core procoagulant role of thrombin is cleavage of fibrinogen into fibrin to generate a thrombus by forming a mesh at the site of infection, …” And at the site of vascular damage?

16) Page 8, lines 362-363: “Thrombin generation can occur as the endpoint of the intrinsic (FXII-mediated) or extrinsic (TF-mediated) pathways of the coagulation cascade.” The intrinsic-extrinsic nomenclature is often replaced with concepts from the cell-based model of haemostasis, as mentioned in one of my previous comments. Again, I think it could be beneficial to introduce the haemostatic system early in the paper.

17) Page 9, lines 393-395: “Administration of an anti-TF monoclonal antibody or the thrombin inhibitor dabigatran etexilate significantly increased CVB3-induced myocarditis [97], indicating that activation of coagulation contributes to the anti-viral response.” So these results indicate that anticoagulant treatment may not be beneficial in viral disease? This is contrary to what you suggest earlier on page 7, lines 263-267.

18) Page 12, line 462: What is meant by “haemostatic coagulation”?

19) Page 13, line 517: “…, and therefore, elevated F3 expression is a clinical biomarker of APS.” I am not sure that measurement of F3 gene expression would a clinically useful biomarker. I also think that, while the section on IFN and thrombosis in APS is interesting, it is important to underline that mechanisms behind thrombosis in APS are probably multifactorial and still far from elucidated.

Reviewer 2 Report

Type 1 interferons (IFNs) are widely expressed cytokines that are readely induced in response to a variety of viral and bacterial infections and shape the antimicrobial innate immune response. The present review deals with the role of IFNs in blood coagulation as modulators of vascular functions and thrombosis. The review covers clearly and extensively different aspects of IFNs as drivers of blood coagulation and the relationship with thromboinflammatory conditions, as well as the possible benefit of targeting IFNs in these clinical scenarios.

Comments

- Clarify the effect of IFNs on platelets and endothelium as contributors to the thrombotic process. In particular, the role of activated platelets to actively release HMGB1 into extracellular space and the IFN/HMGB1/caspase-11 mediated disruption of the endothelial Tie 2 axis

- Emphasize the role of IFNs, not only as regulators of coagulation in myeloproliferative neoplasms, but also as prothrombotic biomarkers (ref 109)

- Stress that the biological funcions of IFNs in host antibacterial immune responses may open new avenues for the prevention or treatment of DIC in Gram-negative bacterial sepsis
